# *Cyathus striatus* Extract Induces Apoptosis in Human Pancreatic Cancer Cells and Inhibits Xenograft Tumor Growth In Vivo

**DOI:** 10.3390/cancers13092017

**Published:** 2021-04-22

**Authors:** Lital Sharvit, Rinat Bar-Shalom, Naiel Azzam, Yaniv Yechiel, Solomon Wasser, Fuad Fares

**Affiliations:** 1Department of Human Biology, Faculty of Natural Sciences, University of Haifa, Haifa 3498838, Israel; lsharvit@univ.haifa.ac.il (L.S.); rbar-shal@univ.haifa.ac.il (R.B.-S.); nazzam3@univ.haifa.ac.il (N.A.); y_yechiel@rambam.health.gov.il (Y.Y.); 2Department of Biotechnology, MIGAL Galilee Research Institute, Kiryat Shmona 11016, Israel; 3Institute of Evolution and Department of Evolutionary and Environmental Biology, Faculty of Natural Sciences, University of Haifa, Haifa 3498838, Israel; spwasser@research.haifa.ac.il

**Keywords:** pancreatic cancer, apoptosis, in vivo, RNAseq, mushroom extract

## Abstract

**Simple Summary:**

The main aim of the present study is to test the effect of *Cyathus striatus* extract on the cell growth of human pancreatic cancer cells in vitro and in vivo. In addition, the effect of the extract on the gene expression was detected. The results indicated that *Cyathus striatus* extract significantly inhibited the cell viability and induced apoptosis. The treatment of xenograft mice harboring human pancreatic cancer cells significantly inhibited tumor growth through the induction of apoptosis. RNAseq experiments revealed the involvement of the MAPK and P53 signaling pathways and pointed toward endoplasmic reticulum stress induced apoptosis. These results may suggest that *Cyathus striatus* extract may contain pro-apoptotic factors that can be identified and used for the treatment of human cancer.

**Abstract:**

Pancreatic cancer is a highly lethal disease with limited options for effective therapy and the lowest survival rate of all cancer forms. Therefore, a new, effective strategy for cancer treatment is in need. Previously, we found that a culture liquid extract of *Cyathus striatus* (CS) has a potent antitumor activity. In the present study, we aimed to investigate the effects of *Cyathus striatus* extract (CSE) on the growth of pancreatic cancer cells, both in vitro and in vivo. The proliferation assay (XTT), cell cycle analysis, Annexin/PI staining and TUNEL assay confirmed the inhibition of cell growth and induction of apoptosis by CSE. A Western blot analysis demonstrated the involvement of both the extrinsic and intrinsic apoptosis pathways. In addition, a RNAseq analysis revealed the involvement of the MAPK and P53 signaling pathways and pointed toward endoplasmic reticulum stress induced apoptosis. The anticancer activity of the CSE was also demonstrated in mice harboring pancreatic cancer cell line-derived tumor xenografts when CSE was given for 5 weeks by weekly IV injections. Our findings suggest that CSE could potentially be useful as a new strategy for treating pancreatic cancer.

## 1. Introduction

Pancreatic cancer is the fourth leading cause of cancer-related deaths in the US and the seventh worldwide, with a 5-year survival rate of 9% [1,2].

The prognosis of the majority of pancreatic cancer patients is very low due to its aggressive nature and the lack of early diagnosis and effective therapy. The median survival of patients diagnosed with the most advanced stage is 4.5 months, increasing to a median survival of only 24.1 months for those diagnosed with early stage disease [3]. Pancreatic cancer responds poorly to most chemotherapeutic agents, and the effectiveness of radiotherapy remains controversial. Several immunotherapeutic strategies have also been tested in patients with pancreatic cancer; however, most of them failed to show any clinically meaningful effects [4]. The lack of response to conventional treatment may be due to the high resilience of pancreas cancer cells to chemotherapy. Hence, the development of new, safe and effective drugs for the treatment of this cancer may help confront this challenging disease. In the past 70 years, out of all anticancer drugs approved for cancer treatment by the FDA, 74.9% can be classified as naturally inspired agents [5,6]. Advances in synthetic chemistry have allowed natural products to serve as leading compounds in the development of new drugs based on their chemical structures, in addition to optimized pharmacological properties. A good example is paclitaxel (Taxol) and its formulations first isolated in 1971 from the Pacific yew used for the treatment of a several types of cancer. This includes pancreatic cancer, ovarian cancer, breast cancer, lung cancer, esophageal cancer, Kaposi sarcoma and cervical cancer. Other drugs used for cancer treatment include etoposide, topotecan, fluorouracil, temozolomide and more [5,6]. Traditionally, a variety of mushrooms have been used in many different cultures for the maintenance of health and in the prevention and treatment of various diseases [7]. *Cyathus striatus* (CS) is a higher Basidiomycetes mushroom, a member of the Nidulariaceae family. CS is known to contain several biologically active compounds. In 1971, it was reported that CS contain indolic substances [8,9] and a complex of diterpenoid antibiotic compounds, *cyathins* [9]. Furthermore, it was found that CS also possess sesquiterpene compounds called schizandronols [10] and several triterpene compounds: glochidone, glochidonol, glochidiol and glochidiol diacetate. Sesquiterpene compounds are a class of natural chemicals that are commonly synthesized by plant species. Certain types of sesquiterpenes have a unique chemical structure that can target a precise protein required for mitosis. Therefore, some sesquiterpenes can induce cell cycle and mitotic arrest [11,12]. Four triterpene compounds, cyathic acid, striatic acid, cyathadonic acid and epistriatic acid, were unknown prior to their isolation from CS [13]. Triterpenes hold numerous biological activities, such as anticancer, anti-inflammatory, antioxidative, antiviral, antibacterial and antifungal properties [14]. Moreover, it was reported that triterpenes can stimulate the immune response by increasing the expression of IL-6 and TNF-α, G2/M cell arrest, apoptosis induction by decreasing the expression of the antiapoptotic protein Bcl-2 and increase the levels of cleaved caspase-9 [15]. These compounds hold important beneficial influences against gastric, hepatic, colorectal, lung and breast cancers [16].

Drug treatments or chemotherapy are crucial parts of PDAC treatment. The standard adjuvant chemotherapy for pancreatic cancer treatment offers a limited median overall survival/disease-free survival, and chemotherapy with or without surgery is part of the standard care. Therefore, there is a need to develop new strategies for PDAC therapy. In the present study, we demonstrated that CSE decreases the viability and growth of human pancreatic cancer cells both in vitro and in vivo. The antiproliferative effect was found to be mediated through the induction of apoptosis via the intrinsic and extrinsic caspase pathways. In addition, a transcriptome analysis following the treatment with CSE suggests the involvement of both p53 and mitogen-activated protein kinase (MAPK) signaling pathways and may point to stress-induced apoptosis as the mechanism leading to cell death.

## 2. Materials and Methods

### 2.1. Preparations of Mushroom Extracts

Mycelial culture of the investigated mushroom, *Cyathus striatus,* was obtained from the culture collection (HAI) of the Institute of Evolution, University of Haifa, Israel. The selected mushroom strain was grown first on a solid medium and then transferred onto submerged conditions, as described before by Zaidman et al. [17,18]. The strain was grown in submerged conditions for 10 days for biomass production. Culture liquid (CL) of the strain was extracted with ethyl acetate )EAC( (SDFCL, Mumbai, India) in the ratio of 1 (CL): 500 mL (EAC), as described before [19]. The extract was dried, diluted with dimethyl sulfoxide (DMSO) (Sigma-Aldrich, St. Louis, MO, USA) to reach a concentration of 50 mg/mL and stored at −20 °C.

### 2.2. Cell Cultures

The human pancreatic cancer cell lines HPAF-II and PL45 (ATCC, Rockville, MD, USA) were maintained in MEM-EAGLE and DMEM mediums, respectively, supplemented with 1% L-glutamine, 10% fetal bovine serum (FBS) and 1% PenStrep (penicillin + streptomycin) (Biological industries, Beit HaEmek, Israel). The HPAF-II cell line was supplemented with an additional 1% sodium pyruvate, and the PL45 cell line was supplemented with an additional 1% L-glutamine. Cells were grown in a humidified incubator at 37 °C with 5% CO_2_ in the air and served twice a week with fresh medium.

### 2.3. XTT Cell Proliferation Assay

Evaluation of the fungal extract effect on cell viability was performed by the XTT assay. HPAF-II and PL45 cells (10^4^) were seeded in 96-well plates and, 24 h later, were treated with several EAC doses: 1, 2.5, 5, 7.5, 10, 15 and 20 µg/mL for a period of 4, 8 and 12 h. The medium was added to the control wells. Following the treatment, the cell viability was determined by the XTT assay (Biological Industries, Beit HaEmek, Israel), according to the manufacturer’s instructions, using an ELISA reader (BioTek). Experiments were repeated 2–5 times independently. Data were presented as the average proliferation percentage of the respective control.

### 2.4. Cell Cycle and Apoptosis Assays

For the cell cycle distribution analysis, HPAF-II and PL45 cells (10^6^) were treated with 5 and 10 µg/mL of CSE for 4 h. At the end of the incubation period, cells were trypsinized and collected with the growth medium, centrifuged, washed with PBS (Biological Industries, Beit HaEmek, Israel) and fixed with 70% ethanol for one hour at −20 °C. Cells were incubated with 0.1% NP-40 in 4 °C for 5 min, followed by 30 min of incubation on ice with 100 µg/mL RNase (Sigma-Aldrich, St. Louis, MO, USA). Finally, 50 µg/mL propidium iodide (PI) (Sigma-Aldrich, St. Louis, MO, USA) was added for 20 min. Cell cycle analysis was carried out by Fluorescence-activated Cell Sorting (FACS) (Becton Dickinson, Franklin Lakes, NJ, USA); 10,000 cells were counted for the control, as well for the treatment groups.

### 2.5. Annexin-V/PI Double Staining

Apoptotic cell death was evaluated and quantified by flow cytometry based on the Annexin-V FITC and PI double-staining kit (Mebcyto^®^ Apoptosis Kit) (MBL, Nagoya, Japan). HPAF-II and PL45 cells (2 × 10^5^) were seeded in 25-cm^2^ flasks. The next day, cells were treated with ascending concentrations of CSE (0, 2.5, 5, 7.5 and 10 µg/mL) for 4 h. Both adherent and floating cells were collected in order to detect early and late apoptotic events. Treated and untreated cells were harvested by trypsinization, washed and suspended in ice-cold PBS. The washed cell pellets were resuspended in ice-cold-binding buffer containing FITC-conjugated Annexin-V and PI. The samples were incubated at room temperature for 15 min in the dark before analysis by FACS, managed with FACSDiva software (Becton Dickinson, Franklin Lakes, NJ, USA).

### 2.6. TUNEL Assay

HPAF-II and PL45 cells were counted (25 × 10^3^ cells/mL) and seeded on chamber slides (Nunc, Roskilde, Denmark). The next day, the growth medium was replaced with a medium containing 10-µg/mL CSE. After 4 h, the cell morphology was examined using DAPI (Vectashield) (Vector Laboratories, Inc., Burlingame, CA, USA) and TUNEL (In Situ Cell Death Detection Kit) (Roche, Mannheim, Germany) staining. The cell growth medium was removed; cells were washed twice with PBS, fixed for 60 min and permeabilized. Subsequently, the cells were incubated with the TUNEL reaction mixture that contains TdT and fluorescein-dUTP for 60 min and washed twice with PBS. DAPI was added on top of TUNEL-treated cells and visualized by fluorescence microscopy.

### 2.7. Western Blotting

Cell lysates from control and CS-treated cells were prepared using a total of 0.3 mL of ice-cold glycerol lyses buffer supplemented with 40 μL, of protease inhibitors and 1 µl/mL of PMSF (Phenylmethylsulfonyl fluoride) (Sigma-Aldrich, St. Louis, MO, USA). Total protein concentrations were determined using a Bio-Rad protein assay (Bio-Rad Laboratories, Inc. Hercules, CA, USA) based on the method of Bradford (1976) using a BSA standard curve (0–35 μg/mL). Equal amounts of proteins (40 mg) were separated by SDS-PAGE with 10–15% polyacrylamide gels and then transferred onto nitrocellulose membranes. To determine the caspase activity, caspase-3-specific antibodies (Abcam, Cambridge, UK), caspase-9 (Cell Signaling Technology, Danvers, MA, USA) and caspase-8 (kindly provided by Prof. Larish, University of Haifa) were used. PARP cleavage was also detected by Western blotting using PARP-specific antibodies (Cell Signaling Technology, MA, USA). β-actin (MP Biomedicals, OH, USA) was used as the loading control. All whole western figures can be found in the Appendix A.

### 2.8. Next-Generation Transcriptome Profiling

HPAF-II and PL45 cells (10^6^ cells seeded in 25-cm^2^ flasks) were treated with a dose of 7.5-µg/mL CSE for 3 h. The total RNA was isolated from the control and treated cells using the mirVana™ miRNA Isolation Kit (Ambion, Inc. Austin, TX, USA), according to the manufacturer’s instructions. Isolated RNA’s concentration and quality was determined by the Qubit^®^ quantitation assay using Qubit^®^ 2.0 fluorometer (Invitrogen, Carlsbad, CA, USA). Samples were prepared for Illumina sequencing using NEB’s Ultra-RNA Library Prep Kit for Illumina (NEB#7530) (New England Biolabs, Ipswich, MA, USA), according to the manufacturer’s protocols. Libraries were loaded at 8 pM and were sequenced with a 50-bp SR run on Illumina HiSeq 2500 using a V3 flow cell.

Adapter sequences were filtered out from Illumina reads, and the fastx tool kit was used to filter out reads with less than 80% of bases under a quality score of 25. The tophat program (v.2.0.6) was used to align reads to the annotated ENSEMBL human genome, allowing 2 mismatches at most for alignment. EdgeR (using default parameters) was used to compare expression levels and call differential expressions, with a q-value of <0.05 required for significance. The transcripts were further filtered at >2-fold changes and >0.05 FDR. 

### 2.9. In Vivo Studies

The in vivo study was approved by the institutional animal experimental ethical committee at the Technion-Israel Institute of Technology (Ethics Number: IL O22-02-11). Eight-week-old athymic nude male mice (Harlan, Indianapolis, IN, USA) were maintained in temperature- and humidity-controlled rooms with a 12-h light/dark cycle and given food and water ad libitum throughout the course of the experiment. A xenograft animal model was generated via the subcutaneous injection of 1 × 10^6^ PL45 cells in the right dorsal flank of nude mice.

Upon tumor formation and growth (50 mm^3^), mice were treated once a week by intravenous (IV) injections with DMSO (*n* = 6 mice) or 2.5-mg/kg CSE (*n* = 8 mice) for 5 weeks. The tumor size was measured biweekly with a caliber, and the volumes were calculated using the formula: length × width^2^ × 0.52. On the final day of the experiment, mice were euthanized using CO_2_, blood samples were obtained by cardiac puncture and plasma was collected by centrifugation and stored at 4 °C for the analysis of the liver and kidney functions. The tumors were removed, weighed, measured and fixed in 10% formalin.

Histological slides of the tumors were prepared by fixation, embedding and section cutting. Apoptotic cells were detected by the TUNEL assay according to the manufacturer’s instructions (Roche Applied Science, Mannheim, Germany).

### 2.10. Statistical Analysis

All experiments, except the in vivo studies, were repeated at least three times (unless indicated otherwise). All data were expressed as a mean value ± standard error (SE), and the statistical differences between groups were evaluated using the Student’s *t*-test for comparison between two groups or ANOVA test (or their nonparametric counterparts) for comparison between multiple groups. *p* < 0.05 was considered statistically significant, and SPSS software was used for the calculation of the differences.

## 3. Results

### 3.1. CSE Inhibits the Growth of Human Pancreatic Cancer Cells

The viability of HPAF-II and PL45 cells treated with different CSE doses for a period of 4, 8 and 12 h was significantly (*p* < 0.001) decreased in a time- and dose-dependent manner (Figure 1A). The viability of cells treated with a dose of 10 µg/mL for 12 h was reduced by over 70%.

In order to clarify the cell viability decrease following CS treatment (2.5–10 µg/mL) for 4 h, a cell cycle distribution analysis was examined. The FACS analysis demonstrated (Figure 1B) that treatment of the cells with CS resulted in accumulation of the cells in the sub-G1 phase of the cell cycle. Up to 70% of the cells were observed in the sub-G1 phase, as indicated by the fluorescent signal. In order to confirm whether this accumulation is a result of apoptosis, HPAF-II and PL45 cells were treated with 10-µg/mL CS for 4 h, stained with FITC-labeled Annexin-V and PI and analyzed by flow cytometry. A significant increase (*p* < 0.05) of Annexin/PI-positive cells (Figure 2A) were captured. The terminal deoxynucleotidyl transferase dUTP nick end-labeling (TUNEL) assay confirmed the presence of apoptotic cells as a consequence of exposure to CSE. As shown in Figure 2B, extensive DNA fragmentation was visible by fluorescence microscopy after TUNEL staining following the treatment compared to untreated cells.

### 3.2. CS Treatment Triggers the Caspase Apoptosis Cascade

To characterize the cell death pathways triggered by CSE treatment, a Western blot analysis was performed to detect the key features of apoptotic cell death involved in both the extrinsic (caspase-8) and the intrinsic (caspase-9) apoptosis pathways. The data presented in Figure 3 demonstrates that the CS treatment of HPAF-II and PL45 cells (2.5–10 µg/mL) for 4 h led to the activation of caspase-8 and caspase-3, followed by the cleavage of Poly (ADP ribose) polymerase (PARP) (Figure 3). Caspase-9 activation was delayed and achieved after 12 h of treatment.

### 3.3. Differential Expression Following Treatment with CS

In order to further investigate the mechanism of CS on the induction of apoptosis in human pancreatic cancer cells, high-throughput next-generation transcriptome profiling was conducted. Changes in the gene expression at the level of mRNA following the treatment were captured. Heat map hierarchical clustering for graphical representation of the data in a matrix that contains all the significantly changed genes (−1 ≤ 1) is presented in Figure 4. The results indicated that 621 genes for the PL45 cell line and 356 genes for the HPAF-II cell line were significantly upregulated, while 166 and 126 were downregulated in PL45 and HPAF-II, respectively. The DAVID Bioinformatic Resources (DAVID) functional annotation analysis revealed 180 gene clusters among the 621 upregulated genes in PL45 cells. Twenty-nine were clustered as positive regulations of apoptotic-related genes with an enrichment score of 8.29 (*p* = 1.3 × 10^−5^ and Bengamini = 6.5 × 10^−4^) (Table 1). For HPAF-II, 105 clusters were found, according to the DAVID functional annotation analysis. Among those clusters, 21 were clustered as positive regulations of apoptotic-related genes with an enrichment score of 5.85 (*p* = 3.8 × 10^−5^ and Bengamini = 3.03 × 10^−3^) (Table 1). Furthermore, for both the PL45 and HPAF-II cell lines, the genes involved in the MAPK signaling pathway were clustered: 22 genes for the PL45 cell line and 11 for the HPAF-II cell line (Table 2). The genes related to the P53 signaling pathway were also clustered for both cell lines: eight genes and six genes for the PL45 and HPAF-II cell lines (Table 3), respectively. Interestingly, none of the downregulated genes in both cell lines revealed any obvious relations to apoptosis, the MAPK signaling pathway or p53 signaling. 

### 3.4. CSE Inhibits the Tumor Growth of Pancreatic Cancer Cells In Vivo

To examine the in vivo antitumor efficacy of CSE, mouse-harboring pancreatic cancer cell line-derived tumor xenografts were used, as described under “Materials and Methods”. Remarkably, five weekly intravenous (IV) injections of 2.5-mg/kg CSE significantly inhibited the tumor growth (Figure 6A). The tumor volume and tumor weights were significantly (*p* < 0.001) decreased as well (data not shown). In addition, paraffin sections of the tumor specimens were stained with DAPI and TUNEL (Figure 6D). The treatment with CSE resulted in the induction of apoptosis in vivo as well. 

In order to reveal any cytotoxic effect of CSE in vivo, the body weights of untreated and treated mice were measured every alternate day. The mice body weights did not vary significantly throughout the study (Figure 6B). Moreover, the results indicated that there was no significant difference in the liver and kidney parameters in the blood samples of the control and treated animals (Table 4).

## 4. Discussion

CSE is a mushroom-derived extract isolated from *Cyathus striatus*. Its anticancer effect was first reported in our laboratory [20,21] and has never been explored by other researcher groups, to this day. Pancreatic cancer is a fatal malignancy with a 5-year survival rate of approximately 10% in the USA [22,23]. Although its diagnosis, management and treatment have advanced over the past decade, very little progress in patient outcomes has been made [23]. There is a great deal of potential in mushroom and mushroom-derived products in finding new strategies for cancer treatments. Medicinal mushrooms are a rich source of bioactive compounds, such as polyphenols, polysaccharides, glucans, terpenoids, steroids, cerebrosides and proteins, which can be used for the treatment of various cancers [24].

In the present study, we extensively profiled the Basidiomycetes mushroom *Cyathus striatus* extract (CSE) effect on the inhibition of growth and the induction of apoptosis in a pancreatic cancer model, both in vitro and in vivo. This extract was carefully chosen out of 31 different extracts, as it was the most effective candidate for the inhibition of human pancreatic cancer cell growth [20,21].

Remarkably, the exposure of human pancreatic cancer cell lines, HPAF-II and PL45, to low doses (10 µg/mL < IC_50_) of CSE for a short period (4 h) is sufficient for achieving a profound antiproliferative effect and induction of apoptosis.

The FACS analysis revealed the accumulation of cells in sub-G1 following exposure to CSE, thereby interpreting DNA fragmentation and damage due to cell death. The Annexin-V/PI staining assay showed that cell death occurred via an apoptotic process.

The induction of apoptosis by CSE via both the intrinsic and extrinsic pathways, as was shown by the activation of both caspase-8 and caspase-9, seems to be the mechanism of action of cell death that leads to a decrease in cell viability.

Several other medicinal mushroom extracts have been shown to induce apoptotic cell death. The alcoholic extract of *Ganoderma lucidum* was shown to inhibit cell proliferation and induce apoptosis in human MCF-7 breast cancer cells [25]. Another example is the antitumor activity of lectin that was isolated from *Agrocybe aegerita*. Lectin was shown to exert antitumor effects on various tumor cell lines, including HeLa, SW480, SGC-7901, MGC80-3, BGC-823, HL-60 and mouse Sarcoma 180 cells, via the induction of apoptosis and DNase activities. Moreover, the proapoptotic activities of the fractions obtained from a hot water extract of *P. ostreatus mycelium* were detected in human colorectal and prostate cancer cells [26,27]. Yang et al., showed antiproliferation and growth inhibition effects in MCF-7 breast cancer cells following the administration of the *Antrodia camphorata* extract through the induction of apoptosis [28].

Most of the anticancer therapies, either using drugs or radiations, kill cancer cells mainly by inducing apoptosis through the activation of caspases [29]. Caspase-8 and -9 are key players in the execution of two major routes of apoptosis: the extrinsic pathway and the intrinsic pathway. Indeed, our results demonstrated an increase in activation of caspase-8 and caspase-9, followed by the cleavage of caspase-3 and PARP after the treatment of human pancreatic cancer cells with CSE.

Among other medicinal mushroom extracts that have been reported by various research groups to possess anticancer properties, several mechanisms of action have been discussed. Aqueous extracts of I’m-Yunity™ isolated from the mushroom *Trametes versicolor* have been reported to inhibit cell proliferation and induce apoptosis in HL-60 and U-937 cells, accompanied by a cell type-dependent disruption of the G_1_/S and G_2_/M phases of cell cycle progression. The growth suppression of treated HL-60 cells is correlated with the downregulation of the retinoblastoma (Rb) protein, downregulation of the antiapoptotic proteins bcl-2 and survivin and increase in the bax and cytochrome C levels, as well as the cleavage of PARP [30]. In another research, the D fraction of the maitake mushroom induced apoptosis through the upregulation of BAK-1 and cytochrome C transcripts [31]. Another study reported that purified water-soluble polysaccharide (AMP) from the fruiting bodies of *Armillaria mellea*, a famous traditional Chinese medicinal and edible fungus, exhibited a potent tumor growth inhibitory effect on A549 cells and induced cell cycle disruption in the G0/G1 phase. This was accompanied by apoptosis and disruption of the mitochondrial membrane potential, which led to cytochrome C release from mitochondria and activation of caspase-9 and -3 [32].

Another interesting part of our study outlined the key player genes involved in the anticancer effect of CSE. The RNAseq analyses following CSE treatment revealed several interesting upregulated genes: HMOX1, COX-2, JUN, DDIT3 and CHOP. The HMOX1 (HO-1) gene, which encodes a highly inducible enzyme that metabolizes heme and thereby protects a variety of cells against oxidative stress and apoptosis, was found to be upregulated by more than 50-fold. Interestingly, several studies demonstrated that the upregulation of HMOX-1 led to the induction of apoptosis. Hamamura et al. demonstrated HMOX-1 upregulation as a part of the oxygen-dependent gene regulation in bortezomib-induced apoptosis [33]. Other research groups showed that HMOX-1 is a negative regulator of growth in the epithelium [34], astroglia [35] and T lymphocytes [36]. Duckers et al. (2001) demonstrated that the upregulation of HMOX-1 decreased cell proliferation [37]. These studies were all in accordance with the findings of our current work. The prostaglandin-endoperoxide synthase 2 (PTGS2), known as cyclooxygenase-2 (COX-2), was also upregulated by more than 20-fold. COX-2 is the key enzyme in prostaglandin biosynthesis and acts as a dioxygenase and as a peroxidase. COX-2 has been demonstrated to be involved in the antiapoptotic processes but may also be constitutively expressed in tumors. The constitutive expression of COX-2 in cells and animal models is associated with tumor cell growth and metastasis, enhanced cellular adhesion and the inhibition of apoptosis [38,39]. Notwithstanding, several other studies have presented evidence that COX-2 can be proapoptotic. Hinz et al. (2004) suggested in their study that endocannabinoid analog R(+)-methanandamide (R(+)-MA) and anandamide cause the apoptotic death of H4 neuroglioma cells by a mechanism involving the de novo expression of COX-2. Moreover, suppression of COX-2 activity by the selective COX-2 inhibitor celecoxib was accompanied by an inhibition of the proapoptotic action of R(+)-MA and anandamide on H4 neuroglioma cells [27]. Lin et al. (2011) suggested that constitutive COX-2 expression supports tumor growth and is antiapoptotic, whereas inducible COX-2 is proapoptotic [40]. In other research, a novel proapoptotic mechanism of cannabidiol involving the initial upregulation of COX-2 and PPAR-γ, and a subsequent nuclear translocation of PPAR-γ by COX-2–dependent PGs, was found [41].

JUN is the third-most upregulated gene following the treatment with CSE. JUN is a putative transforming gene of avian sarcoma virus 17. It encodes a protein that is highly similar to the viral protein, which interacts directly with specific targeting DNA sequences to regulate the gene expression. Heidari et al. (2012) reported that acute lymphoblastic leukemia cells induced with Runx2 transcription factor and c-Jun in parallel with the proapoptotic BIM, followed by dexamethasone treatment, prompted apoptosis. Accordingly, apoptosis was decreased in cells harboring dominant-negative c-Jun but increased in cells with c-Jun overexpression [42]. 

Another interesting upregulated gene, by more than 10-fold, is the DDIT3 gene, which encodes a member of the CCAAT/enhancer-binding protein (C/EBP) family of transcription factors. The protein is activated by endoplasmic reticulum (ER) stress and promotes apoptosis, being one of the ER mechanisms of overcoming stress. The activation of DDIT3 is one way of initiating apoptosis, and the overexpression of CHOP and microinjection of CHOP protein have been reported to lead to cell cycle arrest and/or apoptosis [43,44,45,46,47].

Furthermore, the overexpression of CHOP leads to translocation of the Bax protein from the cytosol to the mitochondria [48], which is crucial for the execution of ER stress-mediated apoptosis.

Our transcriptome analysis also pointed to the involvement of two major signaling pathways: P53 and MAPK. At least seven genes involved in the P53 pathway were upregulated as a result of the CSE treatment, e.g., P21, GADD45, Fas, DR5, NOXA, PUMA and CASP9. In addition, it is known that the p53 protein can functionally interact with MAPK pathways, and many studies have illustrated several routes of interactions between the two pathways. The upregulation of several MAPK-related genes, e.g., FAS, GADD45, JNK and HSP27, has been captured, which may point to the activation of stress-induced apoptosis.

Stress can trigger the p53 phosphorylation by P38, JNK or ERK, followed by p53 response initiation and subsequent cell cycle arrest and apoptosis [49,50,51]. Stress-induced p38 acts as a protein kinase to phosphorylate p53 at various serine residues, which, in turn, activates p53, leading to p53-mediated cellular responses, such as apoptosis. The JNK pathway also involves a phosphorylation cascade that leads to p53 activation [52].

Alongside the molecular evaluation of the effect of CS, a weekly intravenous CSE dose of 2.5 mg/kg was shown to inhibit the growth of tumors in xenografts compared to the control. In addition, TUNEL-positive cells were captured in tumor sections of treated mice, indicating the induction of apoptosis. The liver and kidney enzyme levels confirmed the lack of obvious systemic toxicity in comparison to the control.

Xu et al. 2012 also demonstrated that the extract of *Pleurotus pulmonarius* suppressed the proliferation, invasion and drug resistance of liver cancer cells in vitro and in vivo, mediated by the inhibition of the autocrine VEGF-induced PI3K/AKT signaling pathway [53]. In addition, PSK, protein-bound polysaccharide, derived from Coriolus versicolor and Irofulven (MGI 114), a novel antitumor agent synthesized from the natural product illudin S, as well as sesquiterpene obtained from the mushroom Omphalotus illudens, inhibit tumor growth in vivo [54,55].

## 5. Conclusions

This research demonstrated for the first time a profound anticancer activity of the *Cyathus striatus* extract. CSE suppressed the growth of human pancreatic cancer cells in vitro and in vivo through the induction of apoptosis via caspase-dependent pathways and possibly via ER stress-induced apoptosis and activation of the MAPK and P53 signaling pathways. Future studies will focus on a better understanding of the specific molecular mechanisms underlying this effect. The findings of this research may suggest that CSE could be potentially useful as a new strategy for treating pancreatic cancer.

## Figures and Tables

**Figure 1 cancers-13-02017-f001:**
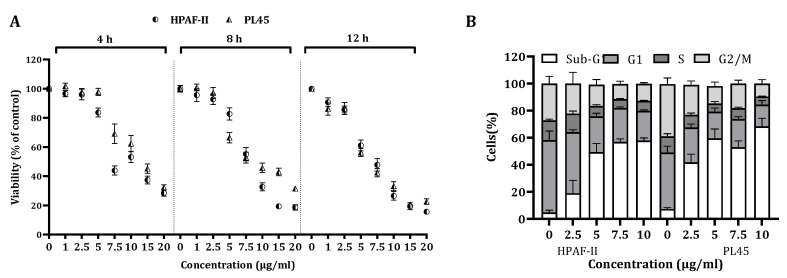
The effect of CS on cell viability and cell cycle progression. (**A**) HPAF-II and PL45 human pancreatic cancer cells were treated with 2.5, 5 and 10 µg/mL for 4, 8 or 12 h, followed by cell viability assessment using the XTT assay, as described under “Materials and Methods”. (**B**). Distribution of HPAF-II and PL45 cells at the sub-G1 phase of the cell cycle following CSE treatment with 2.5, 5 and 10 µg/mL for 4 h. At the end of the treatment, cells were harvested, fixed and stained with PI and subjected to a cell cycle analysis using FACS Calibur (Becton Dickinson, Franklin Lakes, NJ, USA). Statistical significance was determined by one-way ANOVA or the Kruskal–Wallis test, *p* < 0.05, and Mann–Whitney for the post hoc, *p* < 0.007.

**Figure 2 cancers-13-02017-f002:**
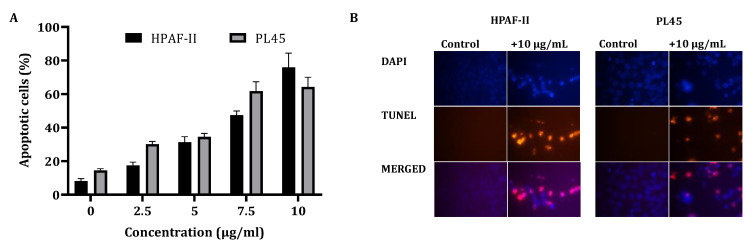
Detection of apoptotic cells using Annexin-V/PI double-staining (**A**) and the TUNEL assay (**B**). (**A**) PL45 and HPAF-II cells were treated with CSE 2.5, 5, 7.5 and 10 µg/mL for 4 h, and a flow cytometric analysis of Annexin V-FITC/PI was performed. The data is presented as the mean (total apoptotic cells) ± SE. Statistical significance was determined by the Kruskal–Wallis test, *p* < 0.05, and Mann–Whitney for the post hoc, *p* < 0.0125. (**B**) PL45 and HPAF cells were treated with 10 µg/mL for 4 h, and the TUNEL assay was performed. Illustrations are representatives of three independent experiments. Blue-colored cells (DAPI) are living cells; red-colored cells (fluorescence-labeled dUTP) are apoptotic cells. Cells were visualized by fluorescence microscopy (original magnification ×20).

**Figure 3 cancers-13-02017-f003:**
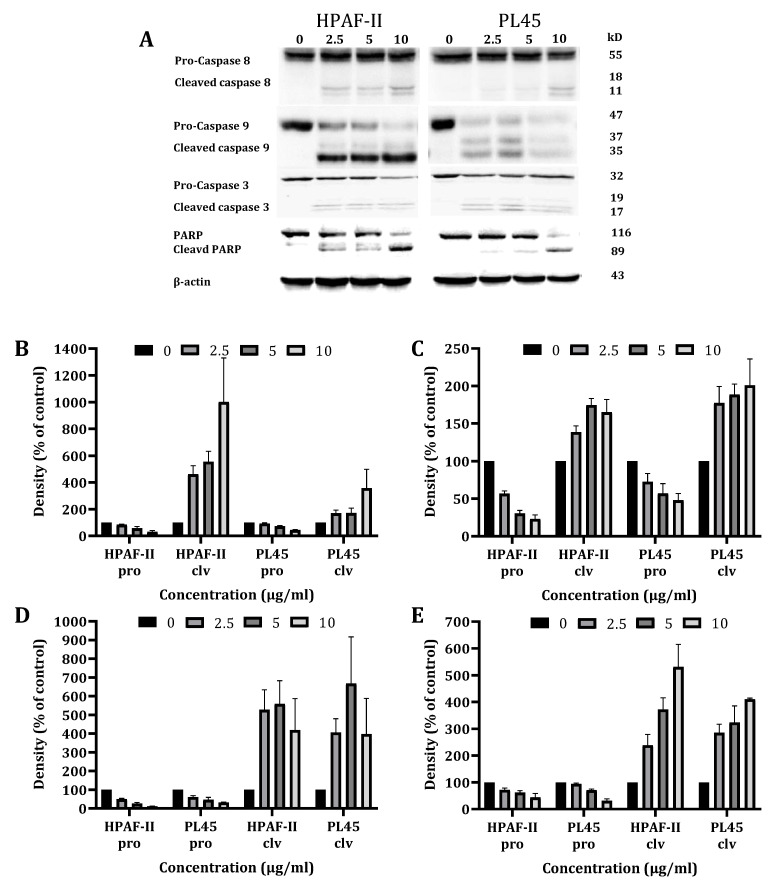
Western blot analysis of the caspases and PARP. HPAF-II and PL45 cells were treated with CSE (2.5, 5 and 10 µg/mL) for 4 h. Representative results (**A**), Specific antibodies for caspase-9 (**B**), caspase-8 (**C**), caspase-3 (**D**) and PARP (**E**) were used. β-Actin was used as the loading control.

**Figure 4 cancers-13-02017-f004:**
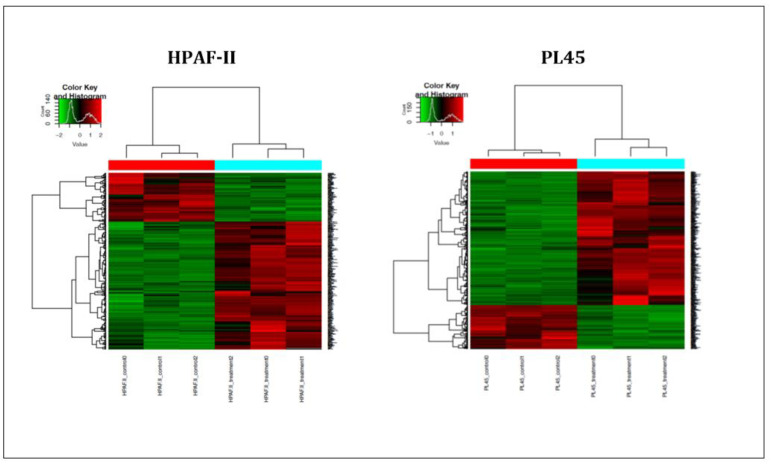
Heat map summarizing the CSE effects on PL45 and HPAF-II. The most significantly down- or upregulated genes are presented. The data is presented in a matrix format, with rows representing individual genes and columns representing each sample. Each cell in the matrix represents the expression level of a gene featured in an individual sample. Red and green reflect the high and low expression levels, respectively, as indicated in the scale bar (log2-transformed scale).

**Figure 5 cancers-13-02017-f005:**
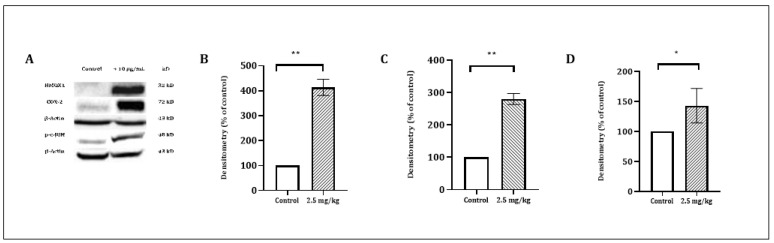
The effect of CSE on the level of COX-2 (72 kD), JUN (48 kD) and HMOX1 (32 kD) proteins. PL45 cells were treated with 10 µg/mL for 4 h. Proteins were extracted and detected by a Western blot analysis, as described under “Materials and Methods”. (**A**) Representative results. (**B**) Quantitative analysis of COX-2 72 kD, (**C**) JUN 48 kD and (**D**) HMOX1 32 kD using Quantity One software. Density values were calculated as a control from the proper β-actin and as a percent of the control. Statistical significance was determined by the Student’s *t*-test (* *p* < 0.01 and ** *p* < 0.001).

**Figure 6 cancers-13-02017-f006:**
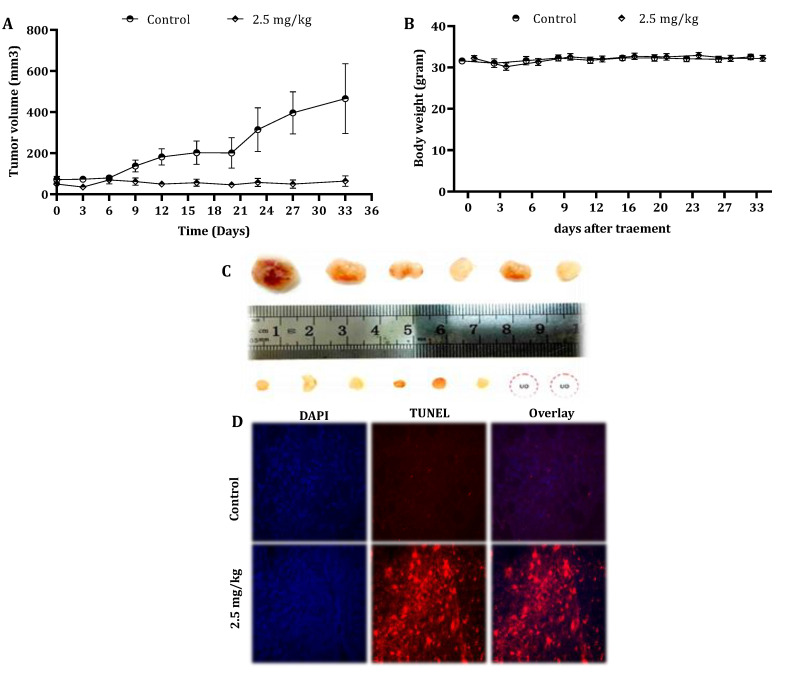
Effect of CSE on the tumor growth of human pancreatic cancer cells in xenograft mice. Nude mice were subcutaneously implanted with 2 × 10^6^ PL45 cells. When the tumors reached a volume of around 50 mm^3^, mice were randomly divided into two groups: control: *n* = 6 and treatment: *n* = 8. The mice were treated once a week by the intravenous injection of either DMSO to the control group or CSE at a concentration of 2.5-mg/kg body weight for 5 weeks. During the experiments, tumor volumes (**A**) and body weights (**B**) were measured twice a week. The results presented are the mean ± SE. Statistical significance was determined by a mixed model repeated-measures ANOVA, *p* < 0.05. Representative tumors from the control and treated mice indicated the inhibition effect of CSE on tumor growth (UO = unobserved) (**C**). Paraffin-embedded tissue sections were prepared from the tumor tissues (control and CS-treated mice), and DAPI and TUNEL staining was performed. Tumor tissues were analyzed under fluorescent microscopy. The TUNEL-positive (orange) cells are apoptotic cells, nuclei are labeled with DAPI (blue) and the merge between DAPI and TUNEL appears pink (magnification, 400×) (**D**).

**Table 1 cancers-13-02017-t001:** Upregulation of apoptosis-related genes in PL45 and HAPF-II cells following the treatment with CSE (FDR-adjusted q-value < 0.05).

Gene Symbol	PL45		HPAF-II	
	logFC	FDR	logFC	FDR
HMOX1	5.76	2.31 × 10^−41^	4.41	7.82 × 10^−14^
PTGS2	4.49	1.31 × 10^−38^	2.24	1.88 × 10^−29^
NR4A1	3.85	1.55 × 10^−85^		
JUN	3.76	8.38 × 10^−79^	2.67	1.75 × 10^−27^
DDIT3	3.40	2.29 × 10^−45^	4.07	7.11 × 10^−41^
CDKN1A	3.08	7.58 × 10^−67^		
DUSP1	3.05	2.45 × 10^−44^	2.15	1.19 × 10^−14^
BCL6	2.94	1.09 × 10^−53^	1.48	1.74 × 10^−10^
CEBPB	2.85	1.46 × 10^−67^	2.49	2.08 × 10^−31^
BBC3	2.72	1.07 × 10^−32^	2.97	4.36 × 10^−14^
DEDD2	2.37	2.61 × 10^−55^	2.08	2.74 × 10^−22^
BCL2L11	2.30	5.12 × 10^−45^		
SOX4	2.04	1.73 × 10^−22^		
UBC	2.01	4.69 × 10^−31^	1.87	3.99 × 10^−23^
IFNB1	1.98	9.72 × 10^−5^		
CASP9	1.76	6.41 × 10^−17^		
CDKN1B	1.67	2.96 × 10^−22^	1.35	6.21 × 10^−8^
SQSTM1	1.63	6.52 × 10^−25^	1.59	1.76 × 10^−5^
BCL3	1.55	1.41 × 10^−9^		
NOTCH1	1.52	2.70 × 10^−20^		
PLEKHF1	1.44	2.68 × 10^−9^		
JMY	1.42	3.55 × 10^−18^	1.74	9.45 × 10^−16^
PMAIP1	1.42	4.64 × 10^−24^	1.58	1.88 × 10^−16^
TNFRSF10B	1.32	2.72 × 10^−21^	1.10	2.47 × 10^−8^
INHBA	1.24	2.07 × 10^−3^		
HSPD1	1.11	4.65 × 10^−11^		
UBB	1.07	4.42 × 10^−15^		
FOXO3	1.04	3.51 × 10^−13^	1.46	3.06 × 10^−12^
PLEKHG2	1.02	6.99 × 10^−6^		
BCL10			1.06	1.78 × 10^−5^
CEBPG			1.11	1.02 × 10^−8^
ERN1			1.70	1.68 × 10^−14^
ING4			1.23	8.57 × 10^−3^
TP53INP1			1.85	9.78 × 10^−6^

**Table 2 cancers-13-02017-t002:** Upregulation of MAPK signaling pathway-related genes in PL45 and HPAF-II cells following the treatment with CSE (FDR-adjusted q-value < 0.05).

Gene Symbol	PL45		HPAF-II	
	logFC	FDR	logFC	FDR
HSPA1A	5.33	1.56 × 10^−149^	4.69	3.45 × 10^−59^
HSPA1B	5.20	6.23 × 10^−129^	3.94	1.90 × 10^−51^
HSPA1L	3.86	2.72 × 10^−19^	2.47	1.06 × 10^−4^
NR4A1	3.85	1.55 × 10^−85^		
JUN	3.76	8.38 × 10^−79^	2.67	1.75 × 10^−27^
DDIT3	3.40	2.29 × 10^−45^	4.07	7.11 × 10^−41^
DUSP1	3.05	2.45 × 10^−44^	2.15	1.19 × 10^−14^
FGFR1	3.03	9.19 × 10^−12^		
JUND	2.42	1.44 × 10^−24^	1.18	4.6 × 10^−4^
MKNK2	2.22	1.67 × 10^−30^		
FGFR3	2.19	7.05 × 10^−7^		
DUSP8	2.17	1.15 × 10^−9^	2.25	5.79 × 10^−16^
FOS	2.13	8.26 × 10^−14^		
DUSP10	2.09	1.58 × 10^−12^	2.68	1.69 × 10^−24^
MAP3K8	2.02	3.32 × 10^−17^		
HSPA6	10.13	7.87 × 10^−67^	7.86	1.54 × 10^−92^
HSPB1	1.65	5.58 × 10^−29^		
HSPA8	1.38	2.25 × 10^−14^		
PLA2G6	1.33	9.33 × 10^−5^		
ACVR1C	1.31	1.7 × 10^−3^		
DUSP2	1.22	5.9 × 10^−4^		
NFATC4	1.14	4.78 × 10^−2^		
DUSP16			1.97	3.69 × 10^−16^
GADD45A			1.30	2.24 × 10^−10^

**Table 3 cancers-13-02017-t003:** Upregulation of P53 signaling pathway-related genes in PL45 and HPAF-II cells following the treatment with CSE. Significantly different control and CS-treated PL45 and HPAF-II cells (FDR-adjusted q-value <0.05). Bold gene names are mutual for both cell lines.

Gene Symbol	PL45		HPAF-II	
	logFC	FDR	logFC	FDR
SESN2	3.47	6.26 × 10^−31^	3.50	4.74 × 10^−30^
CDKN1A	3.08	7.58 × 10^−67^		
BBC3	2.72	1.07 × 10^−32^	2.97	4.36 × 10^−14^
CASP9	1.76	6.41 × 10^−17^		
PMAIP1	1.42	4.64 × 10^−24^	1.58	1.88 × 10^−16^
TNFRSF10B	1.32	2.72 × 10^−21^	1.10	2.47 × 10^−8^
SESN3	1.25	1.20 × 10^−9^		
CCNG2	1.06	4.99 × 10^−7^		
GADD45A			1.30	2.24 × 10^−10^
SIAH1			1.83	8.09 × 10^−23^

Among the most upregulated genes that are related to apoptosis in both cell lines were HMOX1, PTGS2 (COX-2) and JUN. Therefore, the protein expression of these gene products was examined in PL45 cells following 4 h of treatment with CSE. A Western blot analysis revealed a significant increase of these proteins, as presented in Figure 5. These results support the findings of the RNAseq.

**Table 4 cancers-13-02017-t004:** Liver and kidney functions following the treatment with the CS extract. At the end of the treatment period, mice were euthanized, and blood was drawn immediately from the mice hearts. The results presented are the mean ± SE, *n* = 6. Statistical significance was determined by the Student’s *t*-test. Statistical analyses indicated that there was no significant difference between the data from the controls and the treated animals.

	Control	2.5 mg/kg	*p*-Value
Alkalin Phos (U/L)	62.4 ± 4.13	69.1 ± 4.51	0.362
GPT (ALT) (U/L)	41.2 ± 7.03	36 ± 2.62	0.409
GOT (AST) (U/L)	201.8 ± 40.64	189 ± 44.67	0.858
Creatinine J (mg/dL)	0.1975 ± 0.002	0.179 ± 0.01	0.142
UREA (mg/dL)	53 ± 3.65	49.3 ± 2.49	0.435

## Data Availability

The data presented in this study are available on request from the corresponding author.

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
