# Peer review of "Cyathus striatus Extract Induces Apoptosis in Human Pancreatic Cancer Cells and Inhibits Xenograft Tumor Growth In Vivo"

_cancers, 2021, doi:10.3390/cancers13092017_

Round 1

Reviewer 1 Report

The manuscript represents a study with a great methodological design and results of interest of Cyathus striatus Extract Induces Apoptosis in Human Pancreatic Cancer Cells. I believe that research into new therapeutic alternatives is necessary as an adjunctive treatment for cancer. For this reason this manuscript is relevant, however, I have the following comments:

Kerwords: Authors must provide at least 5 keywords.

Introduction

  • Lines 39-42: Indicate the current therapies used in pancreatic cancer. why are they not effective? describe some indicators such as progression free survival, mortality...from clinical trials of
  • Lines 46-51: list some of those drugs that were synthesized in the 1970s. Also describe what did they know for pancreatic cancer?
  • Line 62: “Triterpenes hold nu-62 merous biological activities such as; anti-cancer, Which ones? In what type of cancer? Has action mencanism been studied?
  • the immune response by increasing the expression of IL-6 and TNF-65 α: These actions would not be detrimental to cancer treatment? it would increase inflammation and stimulate tumor cell proliferation. Please clarify
  • They should explain prior to the aim of the study, what is known and why the need for the study.

Material and methods

  • The methodology is well described

Results

  • The results are well described

.Discusion

  • They have included a paragraph on limitations, but it would be appropriate to describe the strengths of this study (of which there are many).
  • Please check the format of the references included in the text.
  • There are many ideas from the discussion not referenced, please include them. Ex: lines 376-380
  • There are many comparisons with other studies, but one could add a reflection of the authors proposing that we are responsible for the reported results.
  • Describe a paragraph with future applications, according to the described mechanism, in which other cancers it could be applied.

Conclusion

  • It is very scattered, please include the main ideas.

Author Response

Answers to the reviewers' comments is attached

Reviewer 2 Report

The authors did not specify any of the components in CSE that promote apoptosis and inhibit tumor growth. This means that the value of clinical use of CSE is very limited. Therefore, it is necessary to define the effects of each of the constituents expected to be included in the CSE on apoptosis and tumor growth.

Although the authors present the results of CSE toxicity analysis through body weight and liver/kidney function analysis in a mouse xenograft experiment, it is required an experiment to measure the cell-cycle progression and apoptosis induction efficacy of CSE in non-cancerous human cells.

The authors propose hypotheses that can explain the phenotypes through RNA-seq analysis. However, it is necessary not only to verify the result of RNA-seq analysis on gene expression, but also to determine (ex, using siRNAs) whether the phenotypes occurred depending on the expression of genes whose expression was verified.

In xenografted tumor tissues, it is necessary to verify the mRNA or protein expression of the genes shown in Fig. 5.

Author Response

Answers to the reviewers' comments are attached 

Round 2

Reviewer 1 Report

The authors have responded to all suggestions.

I have nothing more to add

Author Response

The authors would like to thank the reviewer for the constructive comments that help to revise the manuscript in order to be acceptable for publication

Reviewer 2 Report

For the second comment, authors described their experiments using NHFD cells. However, I cannot find the results in the manuscript. They should present the result in the manuscript or supplementary file

Regarding the first and third comments, the authors said that they are parts of further study and not the scope of the paper. I still think all of my comments are important for a high quality paper. 

Author Response

Firstly, the authors would like to thank the reviewer for the constructive comments that help to revise the manuscript in order to be acceptable for publication.

Regarding the comment using  NHFD cells, we add the results containing the effect of cyathus stiatus on cell viability as a supplementary file.

We are sure that the comments of the reviewer are highly appreciated and lead to high quality manuscript.

The conclusion was revised according to the reviewer comments. We agree with the reviewer that there are still open questions that will be investigated in future studies.